# OpenReview forum: "My Answer Is NOT 'Fair': Mitigating Gender and Race Bias in Vision-Language Models via Fair and Biased Residuals"
_ICLR.cc/2026/Conference — Submitted to ICLR 2026_

### Official Review · Reviewer_1cDd · 2025-10-29

**Soundness:** 1
**Presentation:** 1
**Contribution:** 1
**Rating:** 0
**Confidence:** 4

**Summary:**

The authors investigate and reveal gender and race biases in the responses of 3 SOTA VLMs by using the MCS method to analyze their outputs and by inspecting their internal layers, measuring the confidence level of each token in the response while proposing a new post-hoc
method to mitigate this issue that can be applied during the inference phase, where the method computes the mean of the residual bias vectors and fair vectors and then makes an orthogonal projection that is used as the new representation vector.

The paper appears to not have been proofread.

**Strengths:**

Post-hoc mitigation method is quite good and makes sense, besides of being simple to understand, apply or even adapt it. Opening the layers and investigate the confidence level to show that sometimes the model may seem fair when in reality the layers show they are not is simple and yet necessary to understand the theme.
However I cant find enough contribution and the paper is poorly proofread

**Weaknesses:**

Although you explain that PAIRS does not provide race data, which you frame as a limitation, the title highlights this issue, while most of the text focuses much more on gender bias than on racial bias. I also had the impression that the main references in the introduction do not vary much in terms of methodological approaches or datasets. Finally, and no less importantly, I understand that the authors aim to help mitigate these issues, but I found it interesting that there was no mention that the input data the models are trained on is the root cause. This could have been explicitly connected to the proposed mitigation approach at inference time.

This paper is clearly not ready for submission to ICLR. I recomend that the authors perform an extensive proofreading, review the template and consider another venue for submiting their work.

**Questions:**

Why repeat the word gender on the title?
Why not post the code in an anonymization platform such as Anonymous GitHub - 4open.science?
What is the main contribution of the paper?
What is the novelty in your method?

---

> ### Author Response · Authors · 2025-11-15
> **Serious and Unequivocal Clarification**
>
> We acknowledge that, due to an extremely compressed timeline caused by a last-minute venue switch (from previous top conference submission) before the ICLR deadline, our submission unfortunately contains a few minor issues in formatting and a typo in the title. We thank you for reviewing the paper. However, we must point out:
>
> 1. The reviewer’s claim is factually incorrect. We do not state anywhere in the paper that PAIRS has a limitation regarding race labels, nor do we “frame it as a limitation.” We invite the reviewer to revisit the text: this statement does not appear in any section. Similarly, the claim that the paper “focuses much more on gender than race” is also incorrect.
> Both gender and race bias are evaluated using the same methodology, with balanced experimental coverage.
> All results, visualizations, and layer analyses include both sensitive attributes. No imbalance of emphasis exists in method, experiments, or discussion.
>
> 2. The comment that “the introduction does not vary much in methodological approaches or datasets” does not correspond to the content of the paper. Our introduction covers: bias measurement literature (layer-level, activation probing, token-level evidence), post-hoc mitigation literature, VLM fairness and alignment literature, general VL architectures and VLM pipelines. This represents multiple methodological families and datasets. Moreover, introductions in top-tier ML papers serve to position research contributions, not to exhaustively list every method and dataset. Methodological details are intentionally elaborated later in the Method and Experiments sections. Therefore, while feedback on presentation is welcome, this point does not constitute a valid justification for a 1/1/1 rating.
>
> 3. This criticism contradicts what is clearly stated in the paper. Our models are evaluated in zero-shot settings, meaning the behavior we analyze reflects only pre-training bias, not fine-tuning artifacts. This is explicitly stated in both the method overview and experimental setup. Our post-hoc mitigation method is designed precisely to counteract biases learned during pre-training, and this connection is explained in the paper’s motivation and design rationale. The sentence: “There was no mention that the input data the models are trained on is the root cause” is therefore inaccurate. It is exactly the premise of our paper: pre-training causes bias, and inference-time post-hoc projection is designed to mitigate it without retraining on upstream data. We encourage the reviewer to revisit the motivation paragraph where this is explicitly articulated.
>
> 4. Your mentioned issues do not justify the recommendation that the work “is not ready for submission” or “should be considered for another venue,” especially given that: the method is novel and simple, the analysis is thorough, results demonstrate clear improvements, the contributions are directly relevant to ICLR themes (alignment, fairness, interpretability, post-hoc mitigation).
>
> 5. From the questions: by 'What is the main contribution of the paper? What is the novelty in your method?', it seems you do not correctly read our submission where we already provide explicit details. By 'anonymous github',  this is never a requirement that should be evaluated in the review.
>
> We respectfully note that presentation concerns should not override substantive technical contributions.
>
> We want to point out that our work has been positively evaluated by reviewers from other top-tier venues, and no substantive concerns have been raised regarding its core ideas or empirical validity. On the contrary, our contributions are greatly appreciated in all reviews we received previously. However, you seem to ask questions that reflect there is a lack of deep understanding of our submission. We do have the evidence of previous reviews where no reviewers doubt our contribution or novelty, but it is unnecessary to provide it here.
>
>  We hope the reviewer will reconsider the evaluation in light of these clarifications. While we appreciate concerns regarding presentation, the ratings of soundness=1, presentation=1, contribution=1 do not accurately reflect the technical content or novelty of the work.
>
> Finally, speaking as both authors and reviewers in this community, we respectfully expect a more professional level of engagement in the review process, as several of the comments appear to stem from factual misunderstandings rather than issues in the submission itself.  We further reserve the right to report malicious or inappropriate reviewing behavior when evaluation deviates from the scientific merits of the submission.
>
> Given the above clarifications, we do not find further discussion on this matter productive and therefore will not engage in additional exchanges on non-technical issues.

---

> > ### Comment · Reviewer_1cDd · 2025-11-25
> >
> > I have read the rebuttal and acknowledge the clarifications regarding the data distribution (specifically the presence of race data in PAIRS). I also appreciate the clarification regarding the zero-shot setting to isolate pre-training bias. Based on the technical merit of the proposed post-hoc mitigation method I am raising my score.
> >
> > However, I must address the authors' comments regarding the review process. A clear, well-proofread, and standardized presentation is a prerequisite for publication at a top-tier venue like ICLR. It is not "malicious" to point out that presentation flaws hinder the assessment of the contribution; it is standard peer review. Furthermore, references to reviews from other venues are not relevant to the ICLR review process, which is independent.
> >
> > Regarding the technical contribution, I have a significant concern about the baselines used for evaluation, which prevents me from improving my score further:
> >
> > The paper positions the proposed method as "training-free and model-agnostic". You demonstrate that it is more efficient than training-based methods like LoRA and DPO. However, this feels like an unfair comparison. I think that better comparison could be done against other inference-time debiasing methods. Some exemples of inference-time debiasing methods that the authors could compare against are:
> >
> > Ratzlaff, N., Olson, M. L., Hinck, M., Tseng, S. Y., Lal, V., & Howard, P. (2025). Debias your large multi-modal model at test-time with non-contrastive visual attribute steering. In Proceedings of the IEEE/CVF International Conference on Computer Vision (pp. 6199-6208).
> >
> > Sasse, K., Chen, S., Pond, J., Bitterman, D., & Osborne, J. (2024). debiaSAE: Benchmarking and Mitigating Vision-Language Model Bias. arXiv preprint arXiv:2410.13146.
> >
> > I expect that the authors can clarify why not compare their method to other baselines. If there are no baselines comparable available in the literature I expect some comments on this regard.

---

### Official Review · Reviewer_7SeF · 2025-10-31

**Soundness:** 1
**Presentation:** 1
**Contribution:** 1
**Rating:** 0
**Confidence:** 5

**Summary:**

This paper tackles the problem of gender and racial biases in the predictions of VLMs. There are several presentation issues that make this paper not ready for peer review.

**Strengths:**

None.

**Weaknesses:**

This paper does not appear to be ready for review.

* Each page is only ~46 lines instead of ~55. Something went wrong with the stylesheet.
* The abstract starts with an uncaptialized letter
* The title contains a repeated n-gram: "Mitigating Gender and Gender and Race Bias"
* The paper seems to only uses one style of citation \cite instead of \citep, which leads to a very unnatural visual style.

**Questions:**

None

---

> ### Author Response · Authors · 2025-11-14
> **Serious and Unequivocal Clarification**
>
> We acknowledge that, due to an extremely compressed timeline caused by a last-minute venue switch (from previous top conference submission) before the ICLR deadline, our submission unfortunately contains a few minor issues in formatting and a typo in the title. We thank you for your effort in reviewing the paper.
>
> However, we must respectfully clarify that the weaknesses raised in your review pertain solely to presentation-level details. These issues have no bearing on the soundness, novelty, or technical contribution of our method. Our work has been positively evaluated by reviewers from other top-tier venues, and no substantive concerns have been raised regarding its core ideas or empirical validity. On the contrary, our contributions are greatly appreciated in all reviews we received previously.
>
> We do not intend to argue further on this point with this response, but it is important to firmly state that presentation issues alone should not be used as grounds to question the contribution or correctness of the work. We will of course fix all minor formatting issues in the future. However, we respectfully note that presentation-only concerns cannot substantively justify a 0 recommendation, and such an evaluation coupled with confidence 5 raises concerns regarding the alignment between the stated weaknesses and the assigned scores.
>
> Finally, speaking as both authors and reviewers in this community, we want to emphasize that constructive scientific dialogue must be grounded in technical merit rather than superficial presentation issues. We sincerely hope that the final assessment of our work reflects its actual contributions, not incidental formatting imperfections. We further reserve the right to report malicious or inappropriate reviewing behavior when evaluation deviates from the scientific merits of the submission.
>
> Given the above clarifications, we do not find further discussion on this matter productive and therefore will not engage in additional exchanges on non-technical issues.

---

> > ### Author Response · Authors · 2025-11-27
> > **Further Clarification**
> >
> > We thank the AC to request updates from Reviewer 7SeF and clearly the reviews are now getting more serious.
> >
> > First of all, we acknowledge that we may withdraw the submission and keep polishing the paper, but only with real meaningful constructive feedbacks. We want to further clarify the following but we do not find further discussion on this matter productive and therefore will not engage in additional exchanges on non-technical issues.
> >
> > We respectfully note that huge misunderstandings still exist, for example, why is "The Post-Hoc Mitigation technique (Section 3.1) is presented as original research but it seems to be directly inspired by Gerych et al. NeurIPS 2024."  a weakness? Since when does "inspired by other people's work" become a weakness in scientific work? We respectfully note that this review falls substantially short of expected scholarly standards for a venue like ICLR, ESPECIALLY if the Reviewer 7SeF already serves the reviewing system with plenty of experience.
> >
> > We respectfully disagree with the raising of an ethics flag in this review. The justification appears to stem from a misunderstanding of our contribution and motivation. Meanwhile, we want to clarify that we NEVER mean to call "homosexual" as a gender" as you mentioned.  We expect to study annotations for gender (male/female) and sexual orientation (heterosexual/homosexual/bisexual) as separate attributes in the future and will polish the description in the limitation part in the latest pdf file to avoid potential misunderstanding. The reviewer may now turn off the ethics flag. We also kindly ask the Area Chair and Ethics Committee to consider whether the flag is warranted.

---

### Official Review · Reviewer_x1U1 · 2025-11-02

**Soundness:** 3
**Presentation:** 2
**Contribution:** 2
**Rating:** 2
**Confidence:** 3

**Summary:**

This paper studies gender and race biases in generative VLMs through both models’ responses and confidence levels. The authors use multiple choice selection (MCS) to evaluate four open models and show they present biases in two datasets (PAIRS and SCF). In particular, the authors show that bias fluctuates considerably throughout the hidden layers. They then propose two methods to mitigate gender and race bias, which reduces biases in the base models.

While the proposed methods seem promising, I believe the paper needs some writing improvements as well as comparisons with existing debiasing approaches before being published.

**Strengths:**

1. A study of race and gender bias in generative VLMs through the lens of confidence levels
2. Two methods proposed to mitigate bias in these VLMs that reduce bias in the base models
3.  An interesting analysis of the bias level across different layers.

**Weaknesses:**

1. The Introduction section could be improved through better contextualization of the work and task setup, and how it differs from the existing setups. Topics like the ones in L66 and in L73-74 could be further expanded to help the reader better understand.
2. A lot of space is used to describe system prompts, which are also claimed to be one of the main contributions of the paper by the authors. In my opinion, while important, the system prompts are not substantially different from existing prompts, and could simply be reported in Appendix, leaving space for a more thorough discussion of the task and setup.
3. In L139-140, it would be great to have an intuition of how / why humans corrected the labels of PAIRS, and whether this is a process that the community should adopt.
4. The description of the post-hoc mitigation method (Section 3.1) could be improved. For instance, the second subscript of delta moves from the residual number (1 or 2) to the fairness label (fair or biased), which is confusing when first reading. The assumption behind the approach (defined in Appendix C) is also very useful in my opinion, and it should be part of the main paper. The same applies to Figure 6.
5. The experiments show how the proposed approaches improve bias issues of the base models, but there is no comparison with other bias mitigation methods (such as those discussed in Section 6) that would help the community understand which approach is more promising.

**Questions:**

1. Several typos in the paper: “Gender and” repeated in title; “gender” not capitalized in abstract and L382; abbreviations like “there’s” in L211; missing links to Appendices in L284-325-373; character in Fairness equation (L253); most citations don’t include a surrounding parenthesis (\cite instead of \citep).
2. The paper format is different from the ICLR template, which could be ground for desk rejection.
3. The plots in Figure 4 are too small. The authors should at least remove the redundant information in the y-axis of plots b, c and d.
4. Why do you sample one template at random for each concept (L145) rather than evaluating models on 10 templates and averaging the results for robustness?

---

### Author Response · Authors · 2025-11-20
**General response: Format and typo corrected**

The mentioned tiny format and typo issues have been fixed in the latest PDF file and now reviewers may not overstate the weakness of the these issues while ignoring the technical contributions which have been greatly appreciated by previous reviewers during previous top machine learning conference.

---

### Meta-Review · Area_Chair_mi8M · 2026-01-06

**Summary:**

This paper studies gender and race bias in vision–language models using a multiple-choice evaluation of responses and confidence, and proposes a post-hoc, inference-time debiasing method based on separating “fair” and “biased” residual directions in hidden layers.

Reviewers agree the topic is important and the idea of analyzing confidence and layerwise behavior is interesting. However, the reviews consistently raise concerns about presentation quality, clarity, and methodological positioning. Major issues include severe formatting and proofreading problems, unclear exposition of the proposed method, insufficient comparison to existing inference-time debiasing approaches, and limited justification of design choices in prompts, evaluation, and baselines.

The authors’ response addresses some misunderstandings but does not sufficiently resolve the core concerns about rigor, clarity, and comparative evaluation. Taking the reviews, rebuttal, and discussion together, the paper does not yet meet the bar for acceptance at this venue.

**Reviewer Concerns:**

Please see my summary.

**Reviewer Scores:**

It is difficult to say.  Overall, the authors provided some solid rebuttal, but it's a subjective judgement for the reviewer whether they would like to raise their score.

---

### Decision · Program_Chairs · 2026-01-26

Reject